# Perspectives on the Potential Benefits of Antihypertensive Peptides towards Metabolic Syndrome

**DOI:** 10.3390/ijms21062192

**Published:** 2020-03-22

**Authors:** Forough Jahandideh, Jianping Wu

**Affiliations:** 1Department of Agricultural, Food and Nutritional Science, Faculty of Agricultural, Life and Environmental Sciences, University of Alberta, Edmonton, AB T6G 2P5, Canada; jahandid@ualberta.ca; 2Cardiovascular Research Centre, Faculty of Medicine & Dentistry, University of Alberta, Edmonton, AB T6G 2S2, Canada

**Keywords:** antihypertensive peptides, glucose intolerance, inflammation, insulin resistance, metabolic syndrome, obesity

## Abstract

In addition to the regulation of blood pressure, the renin-angiotensin system (RAS) also plays a key role in the onset and development of insulin resistance, which is central to metabolic syndrome (MetS). Due to the interplay between RAS and insulin resistance, antihypertensive compounds may exert beneficial effects in the management of MetS. Food-derived bioactive peptides with RAS blocking properties can potentially improve adipose tissue dysfunction, glucose intolerance, and insulin resistance involved in the pathogenesis of MetS. This review discusses the pathophysiology of hypertension and the association between RAS and pathogenesis of the MetS. The effects of bioactive peptides with RAS modulating effects on other components of the MetS are discussed. While the in vivo reports on the effectiveness of antihypertensive peptides against MetS are encouraging, the exact mechanism by which these peptides infer their effects on glucose and lipid handling is mostly unknown. Therefore, careful design of experiments along with standardized physiological models to study the effect of antihypertensive peptides on insulin resistance and obesity could help to clarify this relationship.

## 1. Introduction

Metabolic syndrome (MetS) is a cluster of several risk factors for cardiovascular disease (CVD), coronary heart disease, and type-2 diabetes [1]. According to the American Heart Association, MetS is present if three or more of the following criteria are met: waist circumference over 102 or 88 cm for men and women, respectively, blood pressure > 130/85 mmHg, fasting triglyceride (TG) > 150 mg/dL, fasting high-density lipoprotein (HDL) cholesterol < 40 or 50 mg/dL for men and women, respectively, and fasting blood sugar > 100 mg/dL [2]. The hypertension definition is different based on the country and organization. In Canada, high blood pressure is defined as the persistent elevation of systolic/diastolic blood pressure over 135/85 mmHg. Hypertension alone is a major risk factor for developing CVDs affecting one billion people worldwide [3]. 

As a classical feature of the MetS, high blood pressure levels are strongly associated with visceral obesity and insulin resistance, the main pathophysiologic features of MetS [4]. It has been reported that about 50% of patients with essential hypertension are insulin resistant [5,6]. The renin angiotensin system (RAS), well known for its significant role in regulation of blood pressure, also plays a key role in the onset and development of insulin resistance [7]. Hyperinsulinemia exacerbates sequence of events which can finally lead to the development of type-2 diabetes. Individuals with MetS are at twice the risk for developing CVD over the next 5 to 10 years and about five times the risk for type-2 diabetes compared with those without the syndrome [8,9]. Administration of olmesartan, an Ang II receptor blocker, to obese diabetic KKAy mice not only lowers blood pressure but also inhibits adipocyte hypertrophy and reduces inflammation and oxidative stress in these mice [10]. Twenty-six weeks of treatment with valsartan, another Ang II receptor blocker, in subjects with impaired glucose metabolism reduced adipocyte size, improved adipose tissue blood flow, and decreased inflammatory markers’ gene expression [11]. Angiotensin converting enzyme (ACE) inhibitors have also been reported to restore cardiomyocyte contractility, hypoxic preconditioning, and β-adrenergic response impaired in MetS [12,13]. The positive effects of ACE inhibitors on lipid profile and insulin resistance have also been observed in obese pediatric patients with MetS and essential hypertension [14]. ACE inhibition has recently been reported to counteract metabolic cardiomyopathy pathways associated with MetS in LDLR−/−; ob/ob, double knockout mice, while also activating cardioprotective mechanisms [15].

Treatment of MetS requires improvement in lifestyle, engagement in physical activity, and a balanced low-energy diet [4]. Adherence to these lifestyle interventions is generally low and patients often need to take pharmacological treatments. The adverse side-effects associated with synthetic drugs and increasing consumer knowledge about the link between diet and health has spurred the interest in researching and developing functional foods to impart health benefits without the undesirable side effects of synthetic drugs [16].

Food protein-derived bioactive peptides have great potential for the development of functional foods and/or nutraceuticals for the prevention and management of MetS and hypertension [17,18,19]. Bioactive peptides can be released from their parent protein by enzymatic treatments, fermentation, or other processing conditions. Bioactive peptides with ACE inhibitory activity are among the most extensively studied peptides [20,21]. However, there is scant information on the effects of antihypertensive peptides on other components of the MetS. Through examining the pathophysiology of hypertension and the association between RAS overactivity and insulin resistance with MetS, this review discusses the potential benefits of antihypertensive peptides on insulin resistance, adipose tissue dysfunction, inflammation, and impaired glucose handling.

## 2. Pathologies for Hypertension and Metabolic Syndrome

The underlying mechanisms for development of hypertension and MetS are complicated and have not yet been completely defined. Visceral obesity, insulin resistance, sympathetic overactivity, oxidative stress, endothelial dysfunction, activated RAS, and inflammation have been suggested as the potential mechanisms [4,22]. A number of these underlying pathologies, such as hyperactivity of the RAS, inflammation, oxidative stress, and endothelial dysfunction, as the contributors to the onset and long-term persistence of hypertension [23], also associated with MetS, have been discussed in the following sections.

### 2.1. Systemic Renin Angiotensin System (RAS)

Blood pressure is controlled by several different interacting biochemical pathways. Systemic RAS, identified as a hormone-based pathway, plays an important role in regulating arterial pressure. Low blood pressure is one of the stimulants of RAS where it triggers the secretion of enzyme renin from the kidneys to the blood stream. Renin hydrolyzes angiotensinogen secreted from the liver to angiotensin I (Ang I). Angiotensin converting enzyme (ACE) is membrane-bound glycoprotein generated by and resident in the lungs, the endothelial cells of the vasculature, and cell membranes of a number of organs in the body [24]. ACE cleaves the inactive Ang I to the potent vasoconstrictor octapeptide, angiotensin II (Ang II) [25]. Ang II increases blood pressure by various mechanisms including constricting resistance vessels, stimulating aldosterone synthesis, cardiac hypertrophic growth and remodeling, and enhancing sympathetic outflow from the brain [26]. Ang II binds to specific cell surface receptors angiotensin type 1 receptor (AT1R) and angiotensin type 2 receptor (AT2R). Most of the known harmful effects of Ang II are related to AT1R activation. Activation of AT2Rs, present in both endothelial and vascular smooth muscle cells, antagonizes the effects of the AT1Rs by mediating vasodilation, releasing nitric oxide (NO), and inhibiting vascular smooth muscle growth [27]. Ang II may also be degraded to Ang III and IV by the action of aminopeptidase A and M, or Ang 1–7 through the actions of ACE2. ACE2 can also degrade Ang I to Ang 1–9 which can be further degraded by ACE to Ang 1–7. This peptide acts upon MAS receptors and opposes Ang II effects [28,29,30]. The most recent identified member of the vasodilatory axis of the RAS is alamandine. This heptapeptide is generated directly from Ang 1–7 through the action of descarboxylase ACE2 and acts upon MAS-related G-protein–coupled receptor member D. Peptides within the vasodilatory axis of the RAS are involved in vasodilation, blood pressure reduction, antihypertrophy, and antihyperplasia suggesting a plausible antihypertensive role of this axis [31,32].

The cardiovascular risk factors such as hypercholesterolemia and diabetes mellitus are closely related to overexpression and increased activation of AT1Rs in the vascular wall [27]. Ang II facilitates the oxidation and uptake of low-density lipoprotein (LDL) via AT1R [33]. The oxidized LDL further up-regulates AT1R expression [34] showing the interaction between hyperlipidemia and RAS in atherogenesis. The enhanced AT1R expression has been shown in the tissues of hypercholesterolemic animals and humans [35]. The association between RAS and insulin resistance has been discussed later.

### 2.2. Endothelial Dysfunction

Abnormalities in the structure and function of blood vessels in terms of endothelial dysfunction, oxidative stress, and vascular remodeling can lead to the development of hypertension. The endothelium, a monolayer of cells lining the lumen of blood vessels, is the key regulator of vascular homeostasis. The healthy endothelium responds to physical and chemical signals by producing various factors that regulate vascular tone, cellular adhesion, smooth muscle cell proliferation, and vessel wall inflammation [36]. Endothelial cells secrete vasoactive molecules that relax or constrict the vessel. NO, endothelium-derived hyperpolarizing factor, and prostacyclin are the vasodilators while endothelin I, thromboxane, and Ang II are the vasoconstrictor compounds derived from endothelium [36]. A shift in the endothelium’s actions toward impaired vasodilation, increased inflammation and prothrombic conditions is known as endothelial dysfunction [37]. NO plays a key role in maintaining the vascular wall in an inert state by the inhibition of inflammation, cellular proliferation, and thrombosis [23]. NO is generated from L-arginine by the action of endothelial NO synthase (eNOS) in the presence of tetrahydrobiopterin. NO activates soluble guanylate cyclase (sGC) in vascular smooth muscle cells to generate cyclic guanylate monophosphate (GMP). GMP mediates vasodilatation by decreasing smooth muscle tone through a reduction in the concentration of intracellular Ca^2+^ or reduced sensitivity to this ion [38,39]. Reduced NO production or loss of NO bioavailability is one of the key mechanisms involved in endothelial dysfunction [36].

Impaired glucose metabolism, obesity, dyslipidemia, and hypertension are all associated with endothelial dysfunction. Endothelial injury, oxidative stress, inflammation, and disruption of NO function and bioavailability are the mechanisms by which these risk factors affect endothelial function in MetS.

Metabolic actions of insulin on promoting glucose disposal are augmented by vascular actions of insulin on endothelium to stimulate NO production with vasodilatory properties. The pathway involved in the production of NO in the endothelium shares noticeable similarities with the pathways in skeletal muscle promoting glucose uptake [40]. Activation of the phosphatidylinositol 3-kinase (PI3K)-dependent pathways by insulin in endothelial cells leads to the downstream phosphorylation of eNOS and NO release [41]. This further results in the relaxation of resistance vessels and increased blood flow to skeletal muscle which accounts for 25–40% of the increase in insulin-stimulated glucose uptake [40]. Conversely, other distinct nonmetabolic branches of insulin signaling pathways stimulate the endothelial release of endothelin I through Ras/mitogen-activated protein kinase pathway. Insulin resistance is characterized by the impairment of PI3K-dependent signaling pathways without any effect on other insulin dependent pathways. Therefore, the imbalance between the production of NO and secretion of endothelin-1 in endothelium (also characteristics of endothelial dysfunction) lead to decreased blood flow, which worsens insulin resistance. The compensatory hyperinsulinemia usually seen in metabolic insulin resistance states also overdrives the mitogen-activated protein kinase-dependent pathways [41]. This leads to more endothelin I secretion, further exacerbating insulin resistance condition. The reciprocal relationship of insulin resistance and endothelial dysfunction that helps to link cardiovascular and metabolic diseases has been discussed extensively elsewhere [40]. Therapeutic interventions in both animal and human studies have clearly shown this link; improving endothelial function ameliorates insulin resistance, whereas improving insulin sensitivity ameliorates endothelial dysfunction [41].

### 2.3. Oxidative Stress

Oxidative stress, resulting from the excessive and/or dysregulated generation of reactive oxygen or nitrogen species (ROS and RNS, respectively), is a major contributor to disease pathologies [42]. Being continuously produced by different mechanisms in the body [43], ROS/RNS are beneficial in low concentrations by acting as regulatory mediators in cell signaling as well as killing invading microorganisms and damaged cells [44,45]. The imbalance between the production of ROS/RNS and cellular antioxidant capacity leads to development of oxidative stress. Oxidative stress damages nucleic acids, proteins, and unsaturated fatty acids, aggravating cellular damage [46].

Figure 1 shows the interplay between RAS, inflammation, oxidative stress, and endothelial dysfunction. Binding of Ang II to AT1R in vessels activates NADPH oxidase which increases endothelial xanthine oxidase-mediated superoxide production in the endothelium. Increased ROS production leads to eNOS uncoupling [47,48] and impaired vascular relaxation through peroxynitrite (ONOO^−^) generation. Peroxynitrite is a highly reactive molecule causing nitration at the tyrosine residues of various proteins resulting in endothelial inflammation and cell death. Increased formation of peroxynitrite also inactivates prostacyclin synthase, leading to the accumulation of inflammatory and prothrombotic eicosanoids [49].

Hypertension linked to the decreased NO bioavailability and/or plasma FFA-mediated ROS exaggeration is considered a primary risk factor for MetS [50]. Obesity, insulin resistance, hypertension, dyslipidemia, and diabetes have been reported to affect mitochondria’s morphology and oxidative phosphorylation functions. Increased mitochondrial oxidative stress may lead to the induction of mitophagy and apoptosis. Accumulation of plasma free fatty acids (FFAs) in MetS increases ROS production, which disrupts endothelial and smooth muscle cells’ functions [51]. FFAs further trigger ROS production in adipose tissue, leading to the development of a systemic inflammatory state which contributes to obesity-associated vasculopathy and cardiovascular risk [52,53,54]. Reduced mitochondrial antioxidant enzymes’ activity, along with increased markers of protein and lipid oxidation, have been reported in adipose tissues of obese patients [55]. Cholesterol disorders, including the accumulation of free cholesterol, oxidized LDL, and glycated HDL, also impair mitochondrial function and increase the release of pro-inflammatory cytokines which can also lead to endothelial dysfunction [56,57]. Increased oxidative stress in MetS due to enhanced amounts of oxidized LDL, protein carbonyl products, and NADPH oxidase activity leads to higher risk of atherosclerosis and myocardial infarction in MetS patients [58].

In summary, regulation of ROS/RNS levels for maintaining normal physiological functions is critical. Antioxidant therapies have shown potential benefits in minimizing vascular injury and prevention or treatment of hypertension [59,60]. The consumption of dietary antioxidants has been reported to maintain the body’s oxidant/antioxidant status with potential benefits on endothelial function, blood pressure, and lipid profile [61,62,63,64,65,66]. Despite the documented benefits of antioxidants on MetS complications, research on the potential effects of antioxidant peptides on MetS is scarce.

### 2.4. Inflammation

Inflammation, the body’s response to non-lethal injury, is an essential component of the immune response for resisting microbial infection and wound healing. However, excessive and uncontrolled inflammatory changes often lead to the development of vascular disease, atherosclerosis, and its complications. The vascular endothelium plays a key role by acting as a gatekeeper for the extravasation of leukocytes, which is a hallmark of inflammation. The interactions between a variety of inflammatory cells, the extracellular matrix, endothelial cells, and vascular smooth muscle cells are involved in the vascular inflammatory response [67]. When endothelial cells undergo inflammatory activation, endothelial permeability and the expression of adhesion molecules such as vascular cell adhesion molecule-1 (VCAM-1), and intercellular adhesion molecule-1 (ICAM-1) increases. Enhanced adhesion molecules’ expression promotes the adherence of the inflammatory cells followed by their migration across the vascular barrier, and recruitment of additional cytokines and growth factors [68]. Aspirin, the non-steroidal anti-inflammatory drug, is the common therapy devised to target the inflammatory component of CVDs [69]. However, a large part of the population suffers from the side effects of the long-term use of these drugs such as gastric bleeding and ulceration. Hyperglycemia, accumulation of FFA and advanced glycation end products, as well as systemic insulin resistance increase chronic inflammation in MetS [70,71]. The release of leptin, C-reactive protein (CRP), TNF-α, and IL-6 in the vasculature leads to endothelial activation and dysfunction [70,72,73]. TNF-α and IL-6 also increase hepatic lipogenesis and stimulate acute phase response. IL-6 increases plasma plasminogen activator inhibitor-1 (PAI-1), fibrinogen, and CRP levels [74]. Elevated CRP levels are associated with adiposity, hyperinsulinemia, hypertriglyceridemia, low HDL cholesterol, and type-2 diabetes [75]. This association has been shown to be cumulative by increasing the number of the defining features of MetS [76]. The relationship between serum CRP and the presence of MetS, hypertension, and diabetes mellitus has also been recently described [77]. Chronic mild elevation of CRP is also independently predictive of future cardiovascular events [78].

## 3. Renin-Angiotensin System (RAS), Oxidative Stress, and MetS

Insulin resistance, hyperinsulinemia, and blood pressure have been correlated in both obese and lean hypertensive subjects [79]. High baseline and continuously increasing fasting insulin levels have been reported to be independent determinants for the future development of hypertension in a 4-year follow-up study in healthy adults [80]. RAS is well known for its significant role in the regulation of blood pressure in the body. The RAS also plays a key role for the onset and development of insulin resistance [7].

The overproduction of the RAS, including Ang II, activation of the AT1R, and increased production of aldosterone, impairs insulin signaling, further exacerbating insulin resistance [81,82]. An elevated level of plasma Ang II decreases insulin sensitivity due to the formation of ROS [83]. ROS production activates signaling kinases which lead to the inactivation of IRS-1 [82]. Accumulation of visceral fat leads to the development of insulin resistance [84,85] and unregulated lipolysis directly contributing to the cascades of inflammation, dyslipidemia, hypertension, and hyperglycemia [86]. Ang II also retards adipocyte differentiation; thus, dysfunctional adipocytes with more lipolytic activity and pro-inflammatory state would further propagate insulin resistance. Locally produced Ang II can also reduce blood supply to adipose depots, reducing the clearance of released free fatty acids and enhancing local inflammation [7]. On the other hand, insulin resistance also up-regulates RAS components [84,85]; thus, a vicious cycle forms that could explain renal and cardiovascular dysfunctions observed in diabetic individuals (Figure 2). In contrast to Ang II, Ang 1-7 has been reported to exert protective effects against insulin resistance [87,88,89]. Blockade of renin has also been shown to improve measurements of insulin sensitivity in human and animal models. Aliskiren (a renin inhibitor) has been reported to improve hyperglycemia, dyslipidemia, and vascular function in rodent and human studies [90,91,92]. The reported improvements in systemic insulin resistance, insulin signaling and glucose uptake in skeletal muscle by renin inhibition were associated with decreases in the levels of Ang II, aldosterone, AT1R, oxidative stress, and fibrosis. Therefore, it is not known whether renin inhibition advances insulin sensitivity directly or if it does so by affecting downstream components of the RAS. Antihypertensive drugs, such as ACE inhibitors and AT1R blockers, have long been used to prevent cardiovascular and renal damage in diabetic patients [93,94]. In addition to lowering blood pressure, these agents have also been reported extensively to reduce the incidence of new-onset diabetes in patients [95,96,97,98,99]. ACE inhibitors also increase insulin sensitivity by decreasing the formation of Ang II [100]. Valsartan administration in hypertensive, obese individuals with the MetS reduced blood pressure, fasting glucose, and high-sensitivity CRP [101]. The beneficial effects of RAS blockade on MetS and diabetes complications have been attributed to a variety of factors including diminished levels of inflammatory cytokines, elevated adiponectin levels, restored endothelial function, insulin-mediated glucose uptake, pancreatic islets structure and function, and enhanced FFA storage capacity of adipose tissue [102,103,104,105].

RAS components expressed in various tissues in the body including but not limited to the kidneys, heart, brain, and adipose tissue are referred to as the tissue or local RAS. Local RAS involves in physiological (e.g., the role of intrarenal RAS in proper nephrogenesis) and pathophysiological (pathogenesis of hypertension and renal injury) conditions in the body [81]. Studies using models of fetal programming of hypertension have demonstrated that alterations in the intrarenal RAS during fetal development could play an important role in mediating structural changes in kidneys and development of hypertension in later life [106]. Activation of both local and systemic RAS associated with insulin resistance has been reported in many obese individuals [107,108]. Local RAS especially Ang II has also been reported to be involved in dysregulated insulin secretion, adipogenesis, and blood flow to muscle [109]. Local pancreatic RAS is present in humans. RAS components including angiotensinogen, (pro)renin, Ang II, AT1R, and Ang 1-7 have been characterized in human pancreatic islet and/or beta cells. Increased pancreatic RAS activity leads to reduced pancreatic blood flow, diminished insulin secretion, and increased inflammation and pancreatic fibrosis [109]. The role of adipose tissue-specific Ang II in adipocyte differentiation has been reported in the literature [110,111]. Local adipose RAS is believed to increase adipocyte size and differentiation leading to obesity and insulin resistance [109]. Ang II is also present in skeletal myocytes [112]. The effects of Ang II on skeletal muscle glucose uptake is related to alterations in skeletal muscle blood flow and alteration in insulin signaling including reduced IRS1 phosphorylation and AKT activation and insulin stimulated Glut-4 glucose translocation [109].

Pharmacological treatments for RAS blockade are, however, associated with side effects, which has led to a shift toward the development of naturally derived alternatives to synthetic drugs in recent decades. Food proteins containing bioactive peptides with a wide variety of biological activities including antioxidant, antihypertensive, antilipidemic, antimicrobial properties and among others have great potential to be used as alternatives to synthetic drugs [17,18,19,113]. In the next section, we have discussed the potential benefits of peptides with anti-inflammatory and/or antioxidant activities in addition to the RAS modulating effects in the context of MetS.

**Figure 2 ijms-21-02192-f002:**
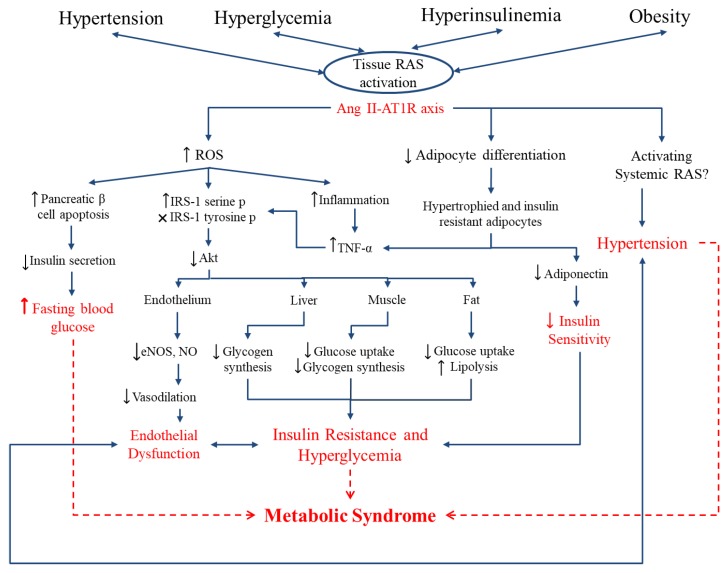
Link between the RAS, oxidative stress, and metabolic syndrome. Hyperglycemia, hypertension, hyperinsulinemia and obesity enhance the expression of local RAS components, especially the Ang II-AT1R axis, in specific tissues. Ang II retards adipocyte differentiation, leading to adipocyte dysfunction and reduced insulin sensitivity through reduced adiponectin secretion and enhanced pro-inflammatory adipokines. Ang II further enhances reactive oxygen species (ROS) production, affecting several pathways in different tissues in the body. Enhanced inflammation, disturbed insulin signaling in endothelium, liver, muscle and adipose tissue leads to endothelial dysfunction and insulin resistance. Furthermore, oxidative stress damages pancreatic β-cells that diminishes insulin secretion and enhances fasting blood glucose. Overall, local RAS overproduction contributes to the enhanced fasting blood glucose, endothelial dysfunction, systemic insulin resistance, and hyperglycemia the hallmarks of metabolic syndrome. Figure is drawn based on [40,114].

## 4. Food Protein Derived Bioactive Peptides; Peptides against RAS and MetS

Bioactive peptides are specific protein fragments that are encrypted in proteins. These peptides can only show their biological effects once they are released from their parent proteins. The diverse physiological effects of bioactive peptides from plant or animal sources have been examined extensively in the literature [115,116,117]. The length of bioactive peptides varies from two (dipeptides) to about twenty amino acid residues [118]. In some cases, much longer peptides with biological activity has also been reported. Lunasin is the peptide derived from soy protein with 43 amino acid residues with anti-cancer and hypocholesterolemic effects [119,120]. Bioactive peptides may influence the major body systems such as cardiovascular, digestive, nervous, and immune systems [121]. The activity is based on their inherent amino acid composition and sequence. Bioactive peptides may have a specific physiological effect or act upon different systems in the body exerting multi-functional physiological properties.

### 4.1. Dairy-Derived Peptides

ACE inhibitory peptides with potential antihypertensive properties have been identified from both animal and plant sources [122,123]. VPP and IPP are the well-known casein-derived tripeptides with antihypertensive, vasculo-protective, antioxidant, and anti-inflammatory properties as shown in vivo [124,125,126,127]. The reported effect of these peptides on reducing blood pressure in clinical trials is controversial [128,129,130,131] and needs further investigation. Potential benefits of protein hydrolysates/bioactive peptides with antihypertensive properties on other complications of the MetS have been summarized in Table 1. Thirty-one-week intake of diets containing VPP and/or IPP including fermented milk, casein hydrolysate and synthesized VPP and IPP attenuated atherosclerosis development in apolipoprotein E–deficient mice [132]. Additionally, VPP and IPP diets reduced mRNA expression of inflammatory cytokines such as IL-6 and IL-1β, and oxidized low-density lipoprotein receptor in this mouse model [132]. Oral administration of VPP for 10 weeks to male C57BL/6J mice at a dosage of 0.3 mg/mL along with a high fat diet exerted anti-inflammatory effects on the adipose tissue of these mice [125]. The adipose tissue of VPP-administered mice showed less activated monocytes and pro-inflammatory macrophages as well as MCP-1 and IL-6 gene expression compared to the control group [125]. Furthermore, these peptides have shown beneficial adipogenic differentiation and insulin mimetic and anti-inflammatory effects in adipocytes, suggesting additional benefits of these peptides in controlling MetS complications [133].

### 4.2. Egg-Derived Peptides

Egg proteins are other sources to produce anti-hypertensive peptides [134,135,136,137,138]. Egg white hydrolysate (EWH) prepared by thermolysin and pepsin reduced blood pressure in spontaneously hypertensive rats (SHRs). This treatment enhanced vasodilation, and improved nitric oxide bioavailability in the vasculature of rats. Interestingly, further mechanistic analysis revealed that egg white hydrolysate modulated the expression level of several components of RAS; reduced ACE and AT1R expression while enhanced AT2R expression in rat’s aorta [137]. Due to the role of Ang II/AT1R axis in the generation of ROS, enhanced NO bioavailability could be the consequence of less oxidative stress in EWH-treated SHRs compared to the untreated control group. EWH treatment also enhanced preadipocyte differentiation and showed insulin mimetic and sensitizing effects in 3T3-F442A preadipocytes [139]. Moreover, EWH treatment improved glucose uptake in TNF-α-treated L6 cells by downregulating the activation of p38-mitogen-activated protein kinase (p38) and c-Jun N-terminal kinases (JNK) 1/2 in these cells [140]. Oral administration of EWH to diet-induced insulin resistant rats also reduced adipocyte size, exerted an anti-inflammatory effect, improved glucose tolerance, and enhanced insulin sensitivity in these rats [141]. WEKAFKDED, QAMPFRVTEQE, ERYPIL, and VFKGL were the bioactive peptides responsible for the observed adipogenic differentiating properties of the EWH [142].

IRW, IQW and LKP are the ovotransferrin-derived tripeptides initially identified as ACE inhibitors [143] and later reported to reduce blood pressure in SHRs [144,145]. IRW and IQW further exerted antioxidant and anti-inflammatory effects in endothelial cells [146,147]. IRW treatment has also been reported to increase the expression and activity of ACE2 in A7r5 cells. ACE2 has a counter-regulatory effect in the RAS by catalyzing Ang II to Angiotensin 1-7, hence exerting vasodilatory effects. When administered to SHRs, IRW treatment upregulated ACE2 protein levels in the kidneys and aorta of these rats [148]. IRW has also been reported to improve insulin resistance in Ang II stimulated L6 cells, at least partially via reduced AT1R expression and its antioxidant activity [149].

GW is a novel di-peptide derived from ovotransferrin hydrolysate with antioxidant and anti-inflammatory effects in human umbilical vein endothelial cells [150]. Both Gly and Trp have been reported as important amino acid residues in exerting anti-inflammatory and antioxidant properties in literature [151,152]. Intraperitoneal injection of Trp to Wistar rats subjected to acute pancreatitis significantly reduced plasma levels of TNF-α while enhanced IL-10 levels with anti-inflammatory properties [153]. Gly supplementation to diets of sucrose fed rats reduced blood pressure, enhanced vascular relaxation, and exerted antioxidant activities in the vasculature of these rats. Enhanced biosynthesis of glutathione by Gly has been reported as the potential mechanism for its antioxidant effect [152]. Lyzozyme is another potential precursor egg protein for the generation of bioactive peptides. Alcalase hydrolysate of lysozyme with ACE-inhibitory activity has been reported to enhance vasorelaxation and reduce inflammation in Zucker diabetic fatty rats [154]. Decreased oxidative stress, inflammation, and COX expression were the mechanisms by which lysozyme-derived peptides attenuated renovascular damage in these rats [154]. Egg white hydrolysates prepared by pepsin and aminopeptidase from Rhizopus oryzae have been reported to exert in vitro ACE-inhibitory activity along with antioxidant and anti-inflammatory effects in the macrophage RAW 264.7 cell line [155]. Administration of pepsin egg white hydrolysate to obese Zucker rats reduced epididymal fat, improved hepatic steatosis, lowered plasma concentration of free fatty acids, and exerted antioxidant and anti-inflammatory effects compared to the control group [156]. Several peptides were identified in pepsin egg white hydrolysate, including FRADHPFL, RADHPFL, YAEERYPIL, YRGGLEPINF, ESIINF, RDILNQ, IVF, YQIGL, SALAM, and FSL [155]. A peptic hydrolysate of egg yolk proteins yielded four peptides; YINQMPQKSRE; YINQMPQKSREA, VTGRFAGHPAAQ, and YIEAVNKVSPRAGQF exerting antioxidant, ACE inhibitory, and antidiabetic (α-glucosidase and DPP-IV inhibitory) activities in vitro [157].

### 4.3. Marine-Derived Peptides

Marine proteins contain peptides with various biological activities including antihypertensive, antiobesity, antioxidant, antimicrobial, immunomodulatory, and anticancer properties [158,159,160]. Pacific cod (Gadus macrocephalus) skin gelatin appears to be a potential source for the production of multifunctional peptides when hydrolyzed by proteolytic enzymes. Hydrolysis with different enzymes yielded different peptides with antioxidant and ACE inhibitory properties. TCSP and TGGGNV were generated when papain was used as the proteolytic enzyme [161], whereas, a longer peptide, LLMLDNDLPP, was generated when gastro-intestinal enzymes were used [162]. MVGSAPGVL and LGPLGHQ were the two ACE inhibitory and antioxidant peptides identified in skate (Okamejei kenojei) gelatin hydrolysate. These peptides showed radical scavenging activity, increased protein level and upregulated gene expression of antioxidant enzymes in human endothelial cells [163]. Sardine protein hydrolysate with ACE inhibitory properties reduced blood glucose levels in stroke-prone SHRs [164]. Animals treated with the sardine protein hydrolysate had lower blood pressure and reduced ACE activity in kidneys, aorta, and mesentery. Administration of the sardine protein hydrolysate also improved glucose tolerance, as measured by an oral glucose tolerance test, without any changes in insulin secretion [164].

MY is an ACE inhibitory dipeptide with antihypertensive effects derived from sardine muscle hydrolysate [165]. This dipeptide also exhibits antioxidant activity by protecting endothelial cells from oxidative stress via induction of heme oxygenase-1 and ferritin [166]. Salmon-skin derived oligopeptides exerted antioxidant and anti-inflammatory effects with the concomitant reduction in fasting blood glucose and protection of β-cells from apoptosis in type 2 diabetic rats [167]. YSQLENEFDR and YIAEDAER isolated from meat and visceral mass extracts of Neptunea arthritica cumingii-a marine snail with high nutritional and commercial value-showed high antioxidant, ACE inhibitory, and antidiabetic activities (α-amylase and α-glucosidase inhibitory activities) in vitro. These peptides also effectively protected skin cells against oxidative damage in a zebrafish model [168]. LSGYGP, a peptide isolated from tilapia skin gelatin hydrolysates, has been recently reported to protect endothelial cells against Ang II-induced injury. LSGYGP inhibited Ang II-stimulated oxidative stress and cytotoxicity, downregulated iNOS and COX-2 by suppressing the NF-κB pathway and enhanced the protein levels of antioxidant enzymes such as superoxide dismutase and glutathione in Ang II-treated human umbilical vein endothelial cells (HUVECs) [169].

### 4.4. Peptides from Other Sources

Hemp seed meal protein hydrolysate has been reported to prevent and treat hypertension in young and adult SHRs, respectively [170]. This treatment also exerted antioxidant activity in vitro and in vivo, further highlighting the interaction of oxidative stress with other pathophysiological conditions and the potential role of antioxidant compounds in such conditions [170].

**Table 1 ijms-21-02192-t001:** Antihypertensive protein hydolysates/peptides with physiological effects on other complications of the metabolic syndrome.

Treatment	Active Component	Model	Observed Effects	Ref
Milk fermented with Lb. case	VPP/IPP	In vitro	Antioxidant activities	[127]
Casein derived peptides	VPP/IPP	L-NAME-treated Wistar rats	Enhanced NO-bioavailability, reduced cardiac and renal damage	[126]
Casein derived peptides	VPP/IPP	In vitro (3T3-F442A preadipocytes)	Adipogenic differentiation, insulin mimetic, and anti-inflammatory effects	[133]
Fermented milk/casein hydrolysate	VPP/IPP	Apolipoprotein E–deficient mice	Reduced mRNA expression of inflammatory cytokines and oxidized LDL- receptor	[132]
Casein derived peptide	VPP	High-fat diet (HFD) fed C57BL/6J mice	Less inflammation in adipose tissue (reduced activated monocytes and pro-inflammatory macrophages, MCP-1 and IL-6 gene expression)	[125]
Egg white hydrolysate	WEKAFKDED, QAMPFRVTEQE, ERYPIL, VFKGL	In vitro (3T3 F442A preadipocytes)	Enhanced preadipocyte differentiation and showed insulin mimetic and sensitizing effects	[139,142]
Egg white hydrolysate	Mixture of peptides	Diet-induced insulin resistant SD rats	Improved glucose tolerance and insulin sensitivity, reduced adipocyte size and inflammation	[141]
Ovotransferrin-derived peptide	IRW	In vitro (HUVECs), SHRs	Reduced inflammatory gene expression, antioxidant and anti-inflammatory effects	[146,171,172]
Ovotransferrin-derived peptide	IRW	In vitro (Ang II-treated L6 cells)	Improved glucose uptake, and antioxidant effects (decreased Ang II-stimulated ROS formation and NADPH oxidase activation)	[149]
Ovotransferrin-derived peptide	IQW	In vitro (HUVECs)	Antioxidant and anti-inflammatory effects	[146]
Lysozyme hydrolysate	Mixture of peptides	Zucker diabetic fatty rats	Decreased oxidative stress, inflammation, and COX expression	[154]
Egg white hydrolysate	FRADHPFL, RADHPFL, YAEERYPIL, YRGGLEPINF, ESIINF, RDILNQ, IVF, YQIGL, SALAM, FSL	Obese Zucker rats	Antioxidant and anti-inflammatory effects, decreased epididymal fat mass, improved hepatic steatosis, and reduced plasma free fatty acids	[156]
Egg white hydrolysate	Mixture of peptides	High-fat/high-dextrose diet-fed Wistar rats	Reduced body weight, abdominal fat, and plasma glucose	[173]
Egg yolk protein hydrolysate	YINQMPQKSRE, YINQMPQKSREA, VTGRFAGHPAAQ, YIEAVNKVSPRAGQF	In vitro	Antioxidant, α-glucosidase and DPP-IV inhibitory activities	[157]
Pacific cod (Gadus macrocephalus) skin gelatin hydrolysate	TCSP, TGGGNV, LLMLDNDLPP	In vitro	Antioxidant	[161,162]
Skate (Okamejei kenojei) gelatin hydrolysate	MVGSAPGVL, LGPLGHQ	In vitro (human endothelial cells)	Antioxidant (radical scavenging activity, increased protein level and upregulated gene expression of antioxidant enzymes)	[163]
Sardine protein hydrolysate	Mixture of peptides	Stroke-prone SHRs	Improved glucose handling and insulin sensitivity	[164]
Sardine muscle hydrolysate	MY	In vitro (human endothelial cells)	Antioxidant activity (protecting endothelial cells from oxidative stress via induction of heme oxygenase-1 and ferritin)	[166]
Wild Chum Salmon protein hydrolysate	Oligopeptides with molecular weights of 130–3000 Da	High fat diet (HFD) fed SD rats	Reduced fasting blood glucose, reduced β-cells apoptosis, antioxidant and anti-inflammatory effects (reduced serum TNFα, IFNγ, and MDA, increased SOD and GSH)	[167]
Marine snail meat and visceral mass	YSQLENEFDR, YIAEDAER	In vitro, and zebrafish model	Antioxidant, α-amylase and α-glucosidase inhibitory activities	[168]
Hemp seed meal protein hydrolysate	Mixture of peptides	SHRs (young and adult)	Antioxidant effects (increased plasma SOD and CAT and decreased total peroxides)	[170]

## 5. Conclusions

A close link between the high prevalence of the non-communicable diseases such as MetS and hypertension globally and dietary habit has prompted interest in unlocking the potential of food bioactives that can be used to improve health or reduce the risk of diseases. One example is bioactive peptides derived from food proteins. Emerging research findings so far have shown the potential effects of antihypertensive peptides in alleviating several complications of the MetS including adipose tissue dysfunction and insulin resistance. Due to the interplay of pathophysiology between the RAS and insulin resistance, there is a great potential for ACE inhibitors to alleviate other complications of the MetS. Understanding the mechanism(s) of action may be challenging due to the multifunctional characteristics of many bioactive peptides where several pathways are involved for their attributed biological effects. Nevertheless, in the case of protein hydrolysates containing a complex array of peptides, assigning the mechanism of action to a single peptide is impossible unless the peptides responsible for that particular biological effect are identified. While the results obtained from these in vitro and in vivo studies are encouraging, studies on humans are scarce. Therefore, clinical trials are necessary to confirm the effectiveness of these peptides and their safety. 

## Figures and Tables

**Figure 1 ijms-21-02192-f001:**
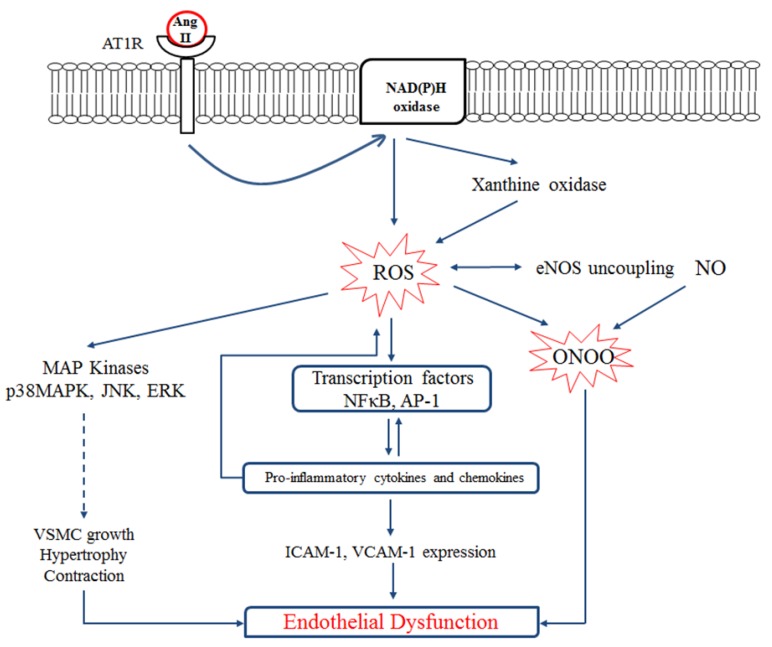
Interplay between the renin-angiotensin system (RAS), oxidative stress, inflammation, and endothelial dysfunction. Ang II induces reactive oxygen species (ROS) production by activating endothelial NAD(P)H oxidase through the AT1 receptor. NAD(P)H oxidase activation further increases endothelial xanthine oxidase-mediated superoxide production and oxidative stress in the endothelium. ROS activates inflammatory transcription factors which enhance expression of adhesion molecules (ICAM-1 and VCAM-1) and/or activates MAP kinases which eventually leads to the VSMC growth and contraction and endothelial dysfunction. On the other hand, ROS induces eNOS uncoupling. NO reacts with superoxide anion resulting in peroxynitrite (ONOO−) formation which also mediates endothelial dysfunction. VSMC, vascular smooth muscle cell; ICAM-1, intracellular cells adhesion molecule 1; VCAM-1, vascular cell adhesion molecule 1. Figure is drawn based on [47].

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
