# Peer review of "Perspectives on the Potential Benefits of Antihypertensive Peptides towards Metabolic Syndrome"

_ijms, 2020, doi:10.3390/ijms21062192_

Round 1
Reviewer 1 Report
The manuscript „Perspectives on the potential benefits of antihypertensive peptides towards metabolic syndrome“ is a review on the pathomechanisms of the individual parts of metabolic syndrome, describes the classical approaches of therapeutic interventions and the potential novel therapy based on food derived bioactive peptides. The potential advantage of this natural-compounds-approach to MetS treatment is aimed to minimize side effects, yet, with maintenance of the adequate protective effect.
In the first part of the review the authors introduce the pathomechanisms of metabolic syndrome along with classical therapeutic interventions. The second part presents the novel, on food derived bioactive peptides with therapeutical potential.
Although the topic is interesting and the work brings a number of interesting data, there is a number of serious concerns, which jeopardize the scientific merit of the manuscript.
-Abstract: Abstract is not written in an optimal way. First, the authors speak about complications of metabolic syndrome- p. 1 line 18- in any case, adipose tissue dysfunction, glucose intolerance or insulin resistance cannot be called complications of MetS - these are individual issues forming together the metabolic syndrome! Complication of metabolic syndrome is considered to be accelerated atherosclerosis and its complications like stroke, myocardial infarction etc., Second, it is not informative sufficiently to say “….these issues are thoroughly discussed in the review” - it would be desirable to name and shortly present the most important compounds and their potential protective effects. Third, the last sentence : …helps to clarify the relationship, if there is any…evokes a doubt, why the review was written- if there is any should be skipped.
- Introduction: is too long. It would be more concise and shorter. Sentence line 72-74 – “…effect of antihypertensive peptides on the complications of MetS – insulin resistance, obesity, lipid tissue dysfunction are not complications of MetS!!! The are the issues of which MetS consists of!
Then several pathomechanisms of metabolic syndrome disturbances are introduced. It is a superficial description of facts, which are known for decades. All this part is too long (page 2 to 7), it should be done more concise and shorter. Moreover, very long sentences repeating the same mechanisms are used…. All issues discussed by the authors are suggested to exert: “…anti-inflammatory, antioxidant, antidiabetic, endothelial protective etc. effects”.and this is repeated many times in many situations almost in the same general way. Such description is rather boring and very general, not being focused on the principle mechanism. Thus, the principle effect of individual pathophysiological or protective factors should be presented, and if there are several of them, it must be a logical sequence in presenting individual mechanisms or effects of a particular pathway.
The main text: Page 3, line 123-124 - today the idea that endothelial dysfunction is principally determined by reduced bioavailability of NO is hardly acceptable, it is more the matter of dysbalance of two groups of compounds with antagonistic actions.
The paragraph page 4, line 155-160 is the oversimplification of reality and should be re-written.
Page 4, line 183- sentences such as “…polyphenols have shown a great potential of alleviating several complications associated with MetS…” represent just mechanical description without any deeper insight and should be avoided or substantially modified.
Several sentences are too long, what could be confusing for readers. It is necessary to split them to smaller parts.
Page 6, 3. Renin angiotensin system: This chapter is based on rather old and widespread data. The difference between local and serum angiotensin II in relation to MetS should be discussed. Moreover, a number of novel issues within RAS have emerged recently such as MAS receptors, alamandine axis, a number of splitting products of Ang II, aldosterone effects etc.- these facts are of value and should be involved.
Moreover, it is rather redundant and inadequate to present such sentence as: p.6, line 249,250 – “…ACE inhibitors (e.g.ramipril, lisinopril, captopril) – or “…AT1R blockers (e.g. valsartan, losartan) – since all these substances are generally well known for al least thirty years and it sounds ridiculously to name some of them.
Page 6, line 251- it is unjustified to name just CAPPP trial as an example of potentially antidiabetic effect of ACE-inhibitors, since there is vast number of similar studies.
Page 6 line 255-266 – this paragraph considering food-derived protectives should not be placed in this chapter, since next chapters are devoted to this particular problem in detail.
Page 7-to 9 – it is an interesting description of food-derived potential protectives. However, the text should be divided into smaller chapters. Additionally, it would be of greater value if the authors would be able to present also the pathomechanisms of action of individual protectives. In this form the chapter has poorly descriptive nature.
References: The references should be completely modified according to the rules for references of IJMS.
Author Response
Response to Reviewer’s Comments (Reviewer 1)
Comments and Suggestions for Authors
The manuscript „Perspectives on the potential benefits of antihypertensive peptides towards metabolic syndrome“ is a review on the pathomechanisms of the individual parts of metabolic syndrome, describes the classical approaches of therapeutic interventions and the potential novel therapy based on food derived bioactive peptides. The potential advantage of this natural-compounds-approach to MetS treatment is aimed to minimize side effects, yet, with maintenance of the adequate protective effect.
In the first part of the review the authors introduce the pathomechanisms of metabolic syndrome along with classical therapeutic interventions. The second part presents the novel, on food derived bioactive peptides with therapeutical potential.
Although the topic is interesting and the work brings a number of interesting data, there is a number of serious concerns, which jeopardize the scientific merit of the manuscript.
-Abstract: Abstract is not written in an optimal way. First, the authors speak about complications of metabolic syndrome- p. 1 line 18- in any case, adipose tissue dysfunction, glucose intolerance or insulin resistance cannot be called complications of MetS -these are individual issues forming together the metabolic syndrome! Complication of metabolic syndrome is considered to be accelerated atherosclerosis and its complications like stroke, myocardial infarction etc.,
Response: The sentence has now been revised for this comment (Line 20).
Second, it is not informative sufficiently to say “…these issues are thoroughly discussed in the review” - it would be desirable to name and shortly present the most important compounds and their potential protective effects.
Response: Since there are different peptides with diverse physiological effects, the authors don’t have the criteria to pick the most important peptides to include in the abstract. Additionally, there is a word limit for the abstract (200 words) as the journal requirement which prohibits authors from presenting any detailed information in the abstract.
Third, the last sentence: …helps to clarify the relationship, if there is any…evokes a doubt, why the review was written- if there is any should be skipped.
Response: The sentence has been modified as per the reviewer’s request i.e. “if there is any” has been deleted (Line 25).
- Introduction: is too long. It would be more concise and shorter. Sentence line 72-74 – “…effect of antihypertensive peptides on the complications of MetS – insulin resistance, obesity, lipid tissue dysfunction are not complications of MetS!!! The are the issues of which MetS consists of!
Response: As per the reviewer’s request, the introduction has been revised for a shorter version yet containing relevant background.
The mentioned sentence has also been modified (Line 69).
Then several pathomechanisms of metabolic syndrome disturbances are introduced. It is a superficial description of facts, which are known for decades. All this part is too long (page 2 to 7), it should be done more concise and shorter. Moreover, very long sentences repeating the same mechanisms are used…. All issues discussed by the authors are suggested to exert: “…anti-inflammatory, antioxidant, antidiabetic, endothelial protective etc. effects”. and this is repeated many times in many situations almost in the same general way. Such description is rather boring and very general, not being focused on the principle mechanism. Thus, the principle effect of individual pathophysiological or protective factors should be presented, and if there are several of them, it must be a logical sequence in presenting individual mechanisms or effects of a particular pathway.
Response: The authors have now revised these sections and made substantial changes to the text. Some sections have now been removed or combined and new information based on the recent literature has been incorporated in the text. The main mechanisms covered in this review paper being RAS, oxidative stress and inflammation related to the pathophysiology of MetS have been introduced and discussed.
The main text: Page 3, line 123-124 - today the idea that endothelial dysfunction is principally determined by reduced bioavailability of NO is hardly acceptable, it is more the matter of dysbalance of two groups of compounds with antagonistic actions.
Response: Endothelial dysfunction is now defined in the text (Lines 121-122), and the above-mentioned sentence is now revised for more clarity and a recent reference is now added to support the sentence (Lines 128-129).
The paragraph page 4, line 155-160 is the oversimplification of reality and should be re-written.
Response: We have made substantial changes in this section (Lines 171-185).
Page 4, line 183- sentences such as “…polyphenols have shown a great potential of alleviating several complications associated with MetS…” represent just mechanical description without any deeper insight and should be avoided or substantially modified.
Response: This paragraph has been deleted.
Several sentences are too long, what could be confusing for readers. It is necessary to split them to smaller parts.
Response: The authors have revised the whole manuscript for more clear and shorter sentences.
Page 6, 3. Renin angiotensin system: This chapter is based on rather old and widespread data. The difference between local and serum angiotensin II in relation to MetS should be discussed. Moreover, a number of novel issues within RAS have emerged recently such as MAS receptors, alamandine axis, a number of splitting products of Ang II, aldosterone effects etc.- these facts are of value and should be involved.
Response: Local RAS was introduced in this section; however, we have expanded this section extensively based on the reviewer’s comment. The differences between systemic and local RAS on MetS have now been included in the text (Lines 238-285). Moreover, the other peptides involved in the RAS are also introduced in section 2.1 (Lines 96-104) for a more comprehensive view as suggested by the reviewer.
Moreover, it is rather redundant and inadequate to present such sentence as: p.6, line 249,250 – “…ACE inhibitors (e.g.ramipril, lisinopril, captopril) – or “…AT1R blockers (e.g. valsartan, losartan) – since all these substances are generally well known for al least thirty years and it sounds ridiculously to name some of them.
Response: The examples of ACE inhibitors and AT1R blockers have now been removed from the text.
Page 6, line 251- it is unjustified to name just CAPPP trial as an example of potentially antidiabetic effect of ACE-inhibitors, since there is vast number of similar studies.
Response: This paragraph has been modified to include more studies and not just one specific trial (Lines 259-260).
Page 6 line 255-266 – this paragraph considering food-derived protectives should not be placed in this chapter, since next chapters are devoted to this particular problem in detail.
Response: Since the next section is on bioactive peptides, this paragraph is the connection between the two sections therefore, we think it is better to keep this paragraph. However, we have revised it for a more concise one (Lines 288-292).
Page 7-to 9 – it is an interesting description of food-derived potential protectives. However, the text should be divided into smaller chapters. Additionally, it would be of greater value if the authors would be able to present also the pathomechanisms of action of individual protectives. In this form the chapter has poorly descriptive nature.
Response: We have now divided this section to group peptides based on their sources. Regarding the comment on the pathomechanisms of action of individual peptides, this information is not available for these peptides to include in the review.
References: The references should be completely modified according to the rules for references of IJMS.
Response: References have now been reformatted based on the journal’s template.
Reviewer 2 Report
The present article gives a comprehensive review focused on the effects of peptides derived from food on oxidative stress, hypertension and metabolic syndrome. The present article also contains a comprehensive review on the pathophysiology related to metabolic syndrome, as a background information for discussing the effects of food-derived peptides. The present review article therefore is useful for multi-disciplinary audience. There are only minor points to be addressed.
1. Please re-check the figures for their accuracy. In particular, Figure 2 contains connecting lines, the purpose of which is unclear. For sample, the origin of the thick dashed blue line pointing to "Tissue RAS activation" is unclear. The direction of the red dashed line connecting between "Fasting blood glucose" and "Metabolic Syndrome complication" is unclear. The overlap of the red dashed line starting from "Insulin Resistance and Hyperglycemia" to the solid blue line coming down from the "Insulin Sensitivity" appears to be messy.
2. Section 4 and Section 5 could be combined. The statement of the current section 4 can be taken as an introductory statement for the detailed contents of the current section 5.
3. Line 33: Criteria of waist circumference should be in SI unit in the scientific writing. Please refer to Grundy et al., Circulation 112: 2735-2752, 2005.
4. Lines 35-37: Please update the criteria of hypertension. It depends on country and organization. In US, therefore according to ACC/AHA, 130 and/or 80 mmHg could be diagnosed as hypertension, Stage 1. In Europe, the criteria of 140/90 is still effective.
5. Line 122-123: The statement that cyclic GMP induced vasodilatation through removal of cytosolic Ca2+ from cell is somewhat misleading. The smooth muscle tone is not only regulated by the cytosolic Ca2+ concentrations but also the by changes in the sensitivity of myofilament to Ca2+. In case of cGMP-mediated vasodilation, decrease in Ca2+ sensitivity plays more important role than decrease in cytosolic Ca2+ concentrations. Please refer to Somlyo & Somlyo, Physiol Rev 83: 1325–1358, 2003.
6. Line 317: "control group" is intuitively misleading. "untreated group" may avoid such misleading.
Author Response
Response to Reviewer’s Comments (Reviewer 2)
Comments and Suggestions for Authors
The present article gives a comprehensive review focused on the effects of peptides derived from food on oxidative stress, hypertension and metabolic syndrome. The present article also contains a comprehensive review on the pathophysiology related to metabolic syndrome, as a background information for discussing the effects of food-derived peptides. The present review article therefore is useful for multi-disciplinary audience. There are only minor points to be addressed.
- Please re-check the figures for their accuracy. In particular, Figure 2 contains connecting lines, the purpose of which is unclear. For sample, the origin of the thick dashed blue line pointing to "Tissue RAS activation" is unclear. The direction of the red dashed line connecting between "Fasting blood glucose" and "Metabolic Syndrome complication" is unclear. The overlap of the red dashed line starting from "Insulin Resistance and Hyperglycemia" to the solid blue line coming down from the "Insulin Sensitivity" appears to be messy.
Response: Figure 2 has now been revised for the issues mentioned by the reviewer.
- Section 4 and Section 5 could be combined. The statement of the current section 4 can be taken as an introductory statement for the detailed contents of the current section 5.
Response: We have combined sections 4 and 5 based on the reviewer’s suggestion.
- Line 33: Criteria of waist circumference should be in SI unit in the scientific writing. Please refer to Grundy et al., Circulation 112: 2735-2752, 2005.
Response: We have the values of waist circumference from inch to cm (Line 32) based on the reviewer’s comment.
- Lines 35-37: Please update the criteria of hypertension. It depends on country and organization. In US, therefore according to ACC/AHA, 130 and/or 80 mmHg could be diagnosed as hypertension, Stage 1. In Europe, the criteria of 140/90 is still effective.
Response: We have updated the criteria of hypertension based on the 2018 Canadian guidelines (the cited reference #3). The sentence has been modified accordingly (Lines 35-37).
- Line 122-123: The statement that cyclic GMP induced vasodilatation through removal of cytosolic Ca2+ from cell is somewhat misleading. The smooth muscle tone is not only regulated by the cytosolic Ca2+ concentrations but also the by changes in the sensitivity of myofilament to Ca2+. In case of cGMP-mediated vasodilation, decrease in Ca2+ sensitivity plays more important role than decrease in cytosolic Ca2+ concentrations. Please refer to Somlyo & Somlyo, Physiol Rev 83: 1325–1358, 2003.
Response: The authors would like to thank the reviewer for raising this point. We have now modified the sentence regarding cGMP-induced vasodilation and have included new references (Lines 126-128).
- Line 317: "control group" is intuitively misleading. "untreated group" may avoid such misleading.
Response: We have changed the “untreated group” to “untreated control group” (Line 344).
Reviewer 3 Report
Dear authors,
Congratulations on the excellent consolidated review on hypertension and its association with insulin resistance. While the review is well written, please make grammatical, typo and punctuational errors in the text.
Author Response
Response: The authors have checked the text thoroughly for these errors and corrected occasional errors in the text.
Round 2
Reviewer 1 Report
The manuscript „Perspectives on the potential benefits of antihypertensive peptides towards metabolic syndrome“ has now been substantially improved.
A number of recommended issues were added and this form of manuscript seems to be a better reflection of the state-of art regarding the renin-angiotensin system.
Hovewer, several minor comments still remain to be addressed:
Line 43-44- please, be sure that other components of metabolic syndrome beside insulin resistance or diabetes cannot be called complications!!! Complication of MetS is atherosclerosis and its complications but not hypertension or inflammation etc… thus: Hyperisulinemia exacerbates sequents of events which can finally lead to the development of type 2 diabetes …
Line 113, 114 - …vessels including endothelial dysfunction … - rather: …of blood vessels in terms of endothelial dysfunction…
Line 171- Hypertension … is a primary tisk factor fot MetS – rather : is primary risk factor of MetS
Line 250-260- the authors used 4 times the same expression – improve or improvement (line 251, 252, 253, 260) – is makes rather awkward impression- please change this expression in some cases
Author Response
Line 43-44- please, be sure that other components of metabolic syndrome beside insulin resistance or diabetes cannot be called complications!!! Complication of MetS is atherosclerosis and its complications but not hypertension or inflammation etc… thus: Hyperisulinemia exacerbates sequents of events which can finally lead to the development of type 2 diabetes …
Response: The sentence has been modified based on the reviewer’s comment.
Line 113, 114 - …vessels including endothelial dysfunction … - rather: …of blood vessels in terms of endothelial dysfunction…
Response: Done.
Line 171- Hypertension … is a primary tisk factor fot MetS – rather : is primary risk factor of MetS
Response: The authors believe that the current form of the sentence is accurate, and the current preposition used (for) does not need to be replaced by “of”.
Line 250-260- the authors used 4 times the same expression – improve or improvement (line 251, 252, 253, 260) – is makes rather awkward impression- please change this expression in some cases
Response: The mentioned sentences have been revised and different expressions have been used to convey the messages.